# An epidemiological model to aid decision-making for COVID-19 control in Sri Lanka

**Dileepa Senajith Ediriweera**[1]*, **Nilanthi Renuka de Silva**[2], **Gathsaurie Neelika Malavige**[3‡], **Hithanadura Janaka de Silva**[4‡]

1 Centre for Health Informatics, Biostatistics and Epidemiology, Faculty of Medicine, University of Kelaniya, Ragama, Sri Lanka, 2 Department of Parasitology, Faculty of Medicine, University of Kelaniya, Ragama, Sri Lanka, 3 Centre for Dengue Research, Faculty of Medical Sciences, University of Sri Jayewardenepura, Nugegoda, Sri Lanka, 4 Department of Medicine, Faculty of Medicine, University of Kelaniya, Ragama, Sri Lanka

☯ These authors contributed equally to this work.
‡ These authors also contributed equally to this work.
* dileepa@kln.ac.lk

**Data Availability Statement:** All relevant data are within the paper and its Supporting Information file.

**Funding:** The author(s) received no specific funding for this work.

## Abstract

### Background

Sri Lanka diagnosed its first local case of COVID-19 on 11 March 2020. The government acted swiftly to contain transmission, with extensive public health measures. At the end of 30 days, Sri Lanka had 197 cases, 54 recovered and 7 deaths; a staged relaxing of the lockdown is now underway. This paper proposes a theoretical basis for estimating the limits within which transmission should be constrained in order to ensure that the case load remains within the capacity of Sri Lanka's health system.

### Methods

We used the Susceptible, Infected, Recovered (SIR) model to explore the number of new infections and estimate ICU bed requirement at different levels of $R_0$ values after lifting lockdown restrictions. We developed a web-based application that enables visualization of cases and ICU bed requirements with time, with adjustable parameters that include: population at risk; number of identified and recovered cases; percentage identified; infectious period; $R_0$ or doubling time; percentage critically ill; available ICU beds; duration of ICU stay; and uncertainty of projection.

### Results

The three-day moving average of the caseload suggested two waves of transmission from Day 0 to 17 (R0 = 3.32, 95% CI 1.85–5.41) and from Day 18–30 (R = 1.25, 95%CI: 0.93–1.63). We estimate that if there are 156 active cases with 91 recovered at the time of lifting lockdown restrictions, and R increases to 1.5 (doubling time 19 days), under the standard parameters for Sri Lanka, the ICU bed capacity of 300 is likely to be saturated by about 100 days, signaled by 18 new infections (95% CI 15–22) on Day 14 after lifting lockdown restrictions.

**Competing interests:** The authors have declared that no competing interests exist.

## Conclusion

Our model suggests that to ensure that the case load remains within the available capacity of the health system after lifting lockdown restrictions, transmission should not exceed R = 1.5. This model and the web-based application may be useful in other low and middle income countries which have similar constraints on health resources.

## Introduction

COVID-19 is caused by a new coronavirus (SARS CoV-2) that emerged in China in December 2019. Although it causes an asymptomatic or mild infection in most instances, it can cause severe respiratory illness or even death. Transmission is mainly via droplets released into the air when an infected person coughs or sneezes. Aerosol and fomite transmission of SARS-CoV-2 is also possible [1, 2]. There is no vaccine at present, nor is there any antiviral agent of proven efficacy. Thus, traditional measures that control the spread of infectious diseases such as quarantine, contact tracing, isolation of positives and contacts as well as social distancing and hand-washing are of vital importance.

The basic reproduction number ($R_0$) is a central concept in infectious disease epidemiology, representing the average number of new infections generated by an infectious person in a completely susceptible population. For COVID-19, $R_0$ has been estimated by the World Health Organization to be 1.4–2.5. Others have placed it higher, at a median of 2.79 with an interquartile (IQR) range of 1.16 [3]. For comparison, seasonal flu has a reported median $R_0$ of 1.28 (IQR, 1.19–1.37), while measles has an $R_0$ of 12–18 [4].

### Situation in Sri Lanka

The 1st case of COVID-19 was diagnosed in Sri Lanka on 27 January 2020, in a tourist from China. The 2nd case was detected nearly 6 weeks later, on 11 March, in a tour guide who probably contracted the infection from Italian tourists. Since then, the spread of infection has been relatively slow, and mostly confined to returnees from countries with high transmission, and their contacts. However, it must be noted that in four of the 190 cases diagnosed in the 30 days from 11 March to 10 April 2020, it was not possible to identify the source of infection. It took nearly a week for the caseload to double from 50 (on 19 March) to 100 on (25 March). It had not yet doubled again as of 11th April, when the count was 197 cases, with 54 recovered and 7 deaths [5]. The epidemic has not yet reached the stage of full-blown community transmission, and almost all cases still occur in clusters where the chain of transmission can be traced.

The government of Sri Lanka acted swiftly to contain transmission, with very stringent public health measures and social distancing: complete island-wide lockdown, contact tracing and isolation, and quarantine of all inbound passengers were all adopted almost simultaneously. The airport has been closed for inbound passengers since 19 March. The national policy with regard to testing was that all symptomatic individuals clinically suspected of infection with SARS-CoV-2, should be tested in one of seven designated laboratories, using PCR as a diagnostic tool. All positive individuals (regardless of severity of illness) are managed in one of three state hospitals, designated for management of COVID-19. These hospitals are also equipped with intensive care units and ventilators for management of the critically ill.

However, the control measures have exacted a very heavy social and economic cost, and the state is now about to implement a phased relaxation of preventive measures. For economic

and social reasons, the government will be forced to re-open Sri Lanka's borders in the near future, while the pandemic is still going on elsewhere.

### Potential impact of COVID-19

It has been suggested that most people infected with SAR-CoV-2 show no symptoms but are still able to infect others. Blanket testing of an isolated village of about 3000 individuals in northern Italy found that 50–75% of infected individuals were asymptomatic [6]. Analysis of the outbreak in China found that 81% of symptomatic individuals had mild illness, whereas 14% developed severe illness (i.e., dyspnea, respiratory frequency $\geq$30/min, blood oxygen saturation $\leq$93%, partial pressure of arterial oxygen to fraction of inspired oxygen ratio <300, and/or lung infiltrates >50% within 24 to 48 h) and another 5% became critically ill with respiratory failure, septic shock, and/or multiple organ dysfunction or failure [7]. It is the provision of effective care for this last group of patients, who may require ventilation for 2–3 weeks, that is the crucial limiting factor in any health system.

The global numbers as of 10 April were 1,617,204 cases, 364,686 recovered, and 97,039 deaths, which suggests a case fatality rate of 5.5% [8]. Of the first 140 patients treated for COVID-19 at the Infectious Disease Hospital in Sri Lanka, where the majority of patients have been managed, nine (6.4%) have required intensive care; a similar proportion to that reported from Wuhan. Sri Lanka's case fatality rate has been 3.7% (7/197) as of 11 April 2020.

If the spread of infection is not controlled, the $R_0$ of SARS-CoV-2 is such that it sweeps swiftly through the susceptible population, resulting in a large number of very ill persons within a short period of time, thus overloading the health system and causing it to collapse. However, it is clearly possible to slow down transmission, as has been demonstrated in Sri Lanka. The availability of beds and ventilators in hospital intensive care units (ICU), to care for critically ill patients, is the major constraining factor that has been observed in all countries with large epidemics. Sri Lanka will need to closely monitor and control the rate of spread of infection so that the requirement for ICU beds and ventilators remains within the available capacity.

This paper proposes a theoretical basis for estimating the limit within which the reproduction number should be constrained, in order to ensure that the infection spreads slowly, and the COVID-19 case load remains within the capacity of Sri Lanka's health system.

### Materials and methods

We used publicly available data for the analysis. The 3-day moving average of cases diagnosed each day during the period 11 March to 15 April were plotted (see Fig 1). These numbers are based on a policy of screening all symptomatic individuals clinically suspected of infection with SARS-CoV-2, using PCR as a diagnostic tool, as recorded in the daily situation reports released by the Epidemiology Unit of the Ministry of Health. It should be noted that an exception to this policy was made on 31 March, when screening was extended to contacts, and 10 of the 21 cases reported on 1 April were asymptomatic positives. Using the maximum likelihood method in the R0 package in R programming language [9], we calculated R over the first 35 days.

We used the Susceptible, Infected, Recovered (SIR) model to explore the number of new infections and estimated ICU bed requirements at different levels of $R_0$ values after lifting lockdown restrictions. These $R_0$ values were selected to represent the range within which transmission may be constrained, and assuming that it will increase after lifting lockdown restrictions (Table 1).

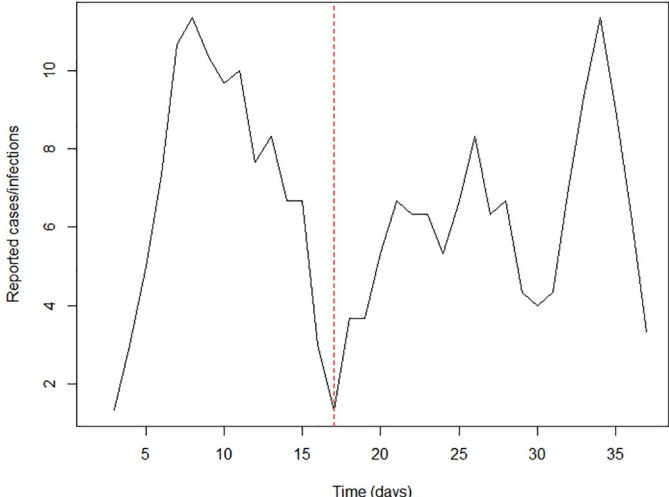

**Fig 1. Three-day moving average of new cases, 11th March to 15th April.** Red dashed line indicate day 17.

We developed a web-based interactive application using an R Shiny package (available through this link: bit.ly/COVID19_ICU) that enables visualization of cases and ICU bed requirements with time under different values of R, with the following adjustable parameters that include the total population at risk; the number of identified cases; the number of recovered cases; the percentage of identified cases among those infected; the infectious period in days; $R_0$ or doubling time in days; the percentage who are expected to become critically ill; the available number of ICU beds; the average duration of ICU stay in days; and uncertainty of projection (coefficient of variation).

We calculated the scenarios that emerge at different values of R, in terms of active infections and ICU requirements subsequent to lifting lockdown restrictions on 20 April 2020, under the following assumptions:

1. the entire population of Sri Lanka (22 million) is susceptible to infection,

2. there are 156 identified cases, and 91 recovered (as recorded by the Epidemiology Unit of the Ministry of Health on 19 April),

3. 50% of all infections are asymptomatic or pre-symptomatic and therefore undiagnosed,

4. the average infectious period is 14 days,

5. 5% of symptomatic patients will require ICU care,

6. the average duration of ICU stay is 2 weeks,

7. maximum critical care capacity = 300 ICU beds and ventilators

**Table 1. $R_0$ values and doubling time of infections.**

| $R_0$ | Doubling time of active infections |
|---|---|
| 1.3 | 32 days |
| 1.4 | 24 days |
| 1.5 | 19 days |
| 1.6 | 16 days |
| 1.7 | 14 days |
| 1.8 | 12 days |

At present, the state hospitals in Sri Lanka have a total of about 670 functional ICU beds with ventilators. While retaining capacity for management of patients with other illnesses, we assumed that up to 300 of these ICU beds may be made available for management of COVID-19 patients at the peak of the epidemic.

## Results

The three-day moving average of daily new cases over the first month (Fig 1) is suggestive of two waves of transmission, and so we calculated R separately for these two periods. The first wave, from Day 0 to Day 17 was largely due to infections among foreign returnees ($R_0$ = 3.32 [95%CI: 1.85–5.41]). The second wave was largely due to local transmission among their contacts ($R_2$ = 1.25 [95%CI: 0.93–1.63]).

Fig 2 shows the possible course of the epidemic if transmission remained at the initial level seen during the first wave of transmission (R = 3.32). This model suggests that the epidemic would have peaked in about 3 months, with more than 5,000,000 affected individuals at the peak of the epidemic.

Fig 3 shows the interface of the web-based application. This web-based application plots the expected epidemic curve under the user input parameters and provides projections on expected new infections on day 7, 14, 21 and 30, new infections by day 7, 14,21 and 30, day of the peak epidemic, infected patients at peak, critically ill patients at peak, required ICU bed days at peak and the day of the ICU saturation under each scenario.

Fig 4 shows how the spread of infection could progress at each of the selected levels of $R_0$. It can be seen that as the value of $R_0$ decreases, the curve becomes flatter: the peak arrives progressively later, and affects a smaller number of individuals at any one time.

Fig 5 shows how saturation of ICU bed capacity (300 beds) could be delayed, as the value of R becomes lower. The curves suggest that while saturation of ICU bed capacity would not occur until about 6 months have elapsed at the lowest value of R selected (R = 1.3), this would happen in about 2 months if R = 1.8, the highest value selected.

We then estimate active infections predicted on Days 7, 14, 21 and 30 after lifting lockdown restrictions and the day of ICU saturation (300 beds) at different values of R (see Table 2), and the predicted new infections over this same period (see Table 3). This suggests

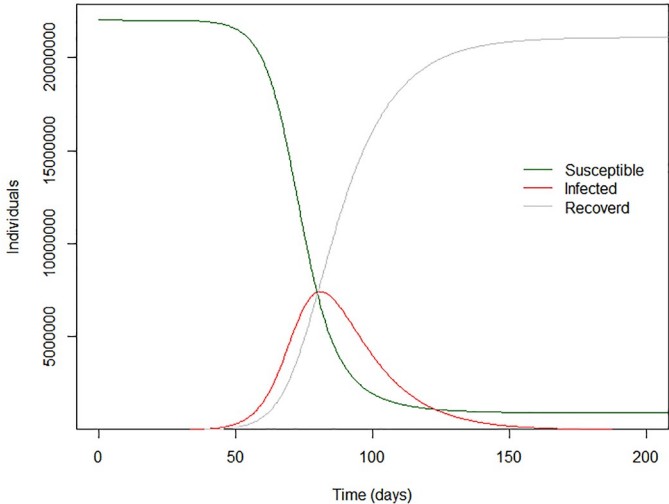

**Fig 2. Natural progression of COVID-19 epidemic when R = 3.32.**

## COVID 19 - Critical care requirement projection

Please see below for caveats and remarks*

### Input parameters:

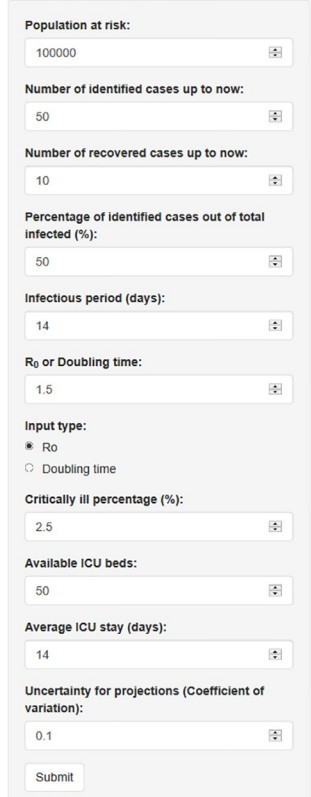

**Population at risk:**

100000

**Number of identified cases up to now:**

50

**Number of recovered cases up to now:**

**Percentage of identified cases out of total infected (%):**

50

**Infectious period (days):**

14

**$R_0$ or Doubling time:**

1.5

**Input type:**
- Ro
- Doubling time

**Critically ill percentage (%):**

2.5

**Available ICU beds:**

50

**Average ICU stay (days):**

14

**Uncertainty for projections (Coefficient of variation):**

0.1

Submit

### Expected Epidemic Curve:

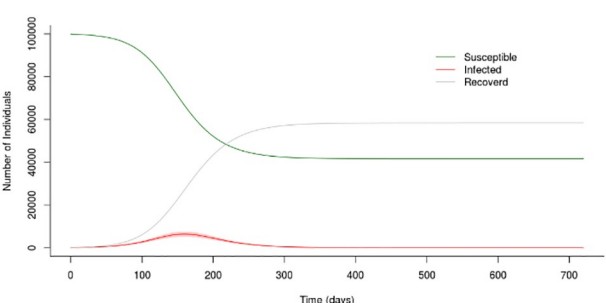

### Projections:

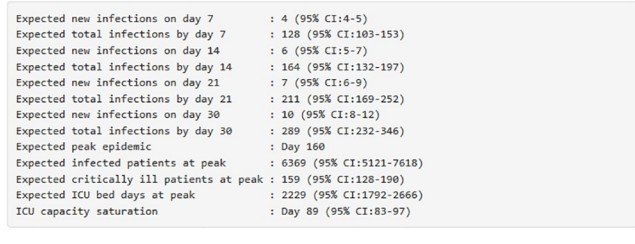

```
Expected new infections on day 7        : 4 (95% CI:4-5)
Expected total infections by day 7      : 128 (95% CI:103-153)
Expected new infections on day 14       : 6 (95% CI:5-7)
Expected total infections by day 14     : 164 (95% CI:132-197)
Expected new infections on day 21       : 7 (95% CI:6-9)
Expected total infections by day 21     : 211 (95% CI:169-252)
Expected new infections on day 30       : 10 (95% CI:8-12)
Expected total infections by day 30     : 289 (95% CI:232-346)
Expected peak epidemic                  : Day 160
Expected infected patients at peak      : 6369 (95% CI:5121-7618)
Expected critically ill patients at peak : 159 (95% CI:128-190)
Expected ICU bed days at peak           : 2229 (95% CI:1792-2666)
ICU capacity saturation                 : Day 89 (95% CI:83-97)
```

**Fig 3. Interface of the web-based application.** URL: bit.ly/COVID19_ICU.

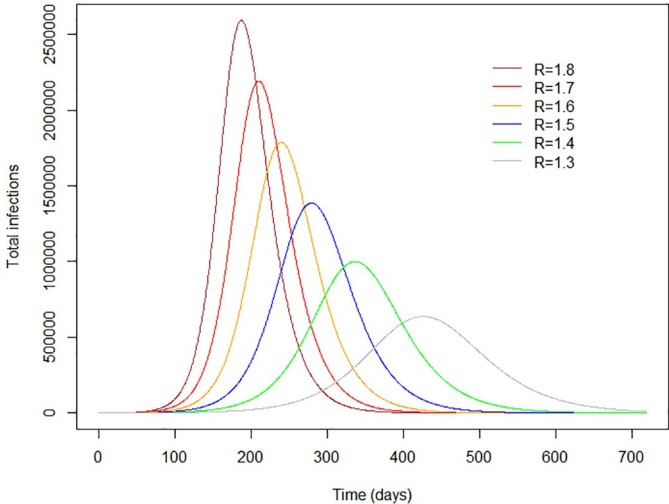

**Fig 4. The epidemic curve over time at selected values of R.**

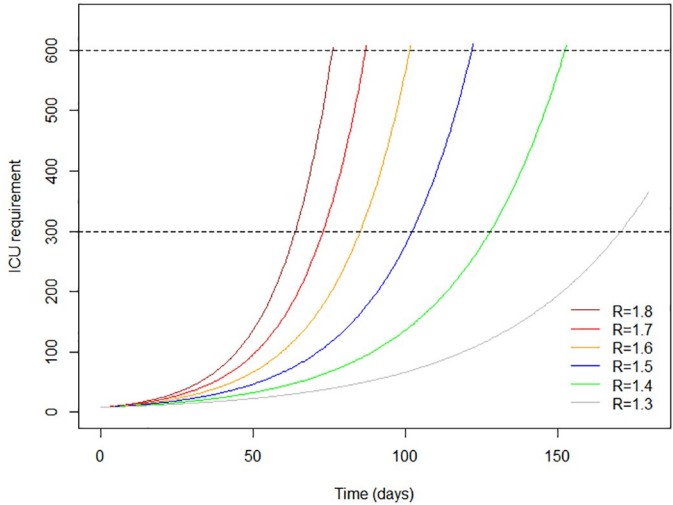

**Fig 5. Saturation of ICU bed capacity: Changes with time at selected values of R.**

**Table 2. Predicted active infections and ICU bed saturation at selected values of R after lifting lockdown restrictions.**

| R value | Active infections after 7 days | Active infections after 14 days | Active infections after 21 days | Active infections after 30 days | ICU saturation on day |
|---|---|---|---|---|---|
| (Day of the expected peak) | (95% CI) | (95% CI) | (95% CI) | (95% CI) | (95% CI) |
| 1.3 (Day 426) | 363 | 421 | 489 | 594 | 171 |
| | (291–434) | (339–504) | (394–585) | (477–710) | (163–181) |
| 1.4 (Day 337) | 381 | 466 | 569 | 736 | 128 |
| | (306–456) | (374–557) | (457–680) | (591–880) | (122–136) |
| 1.5 (Day 280) | 401 | 515 | 661 | 911 | 103 |
| | (322–479) | (414–615) | (531–790) | (733–1090) | (98–109) |
| 1.6 (Day 240) | 421 | 569 | 768 | 1129 | 86 |
| | (339–504) | (457–680) | (617–918) | (908–1351) | (81–91) |
| 1.7 (Day 210) | 443 | 629 | 892 | 1399 | 74 |
| | (356–530) | (505–752) | (717–1067) | (1125–1674) | (70–78) |
| 1.8 (Day 187) | 466 | 695 | 1037 | 1734 | 64 |
| | (374–557) | (559–831) | (833–1240) | (1394–2074) | (61–68) |

**Table 3. Expected new infections on day 7, 14, 21 and 30 at selected values of R.**

| R value | New infections on day 7 | New infections on day 14 | New infections on day 21 | New infections on day 30 |
|---|---|---|---|---|
| (Day of the expected peak) | (95% CI) | (95% CI) | (95% CI) | (95% CI) |
| 1.3 (Day 426) | 8 (6–9) | 9 (7–11) | 10 (8–12) | 13 (10–15) |
| 1.4 (Day 337) | 11 (9–13) | 13 (10–16) | 16 (13–19) | 21 (17–25) |
| 1.5 (Day 280) | 14 (11–17) | 18 (14–22) | 23 (19–28) | 32 (26–38) |
| 1.6 (Day 240) | 18 (14–21) | 24 (19–29) | 32 (26–38) | 47 (38–57) |
| 1.7 (Day 210) | 22 (17–26) | 31 (25–37) | 44 (35–52) | 68 (55–82) |
| 1.8 (Day 187) | 26 (21–31) | 39 (31–46) | 58 (46–69) | 96 (78–115) |

that a R value of 1.5 or above would result in saturation of ICU capacity within about 3 months of lifting lockdown restrictions, and this would be likely if the number of active infections reaches 515 (95% CI 414–615) and approximately 18 (95% CI 14–22) new infections on Day 14 after lifting lockdown restrictions. Based on our assumption that 50% of infections are asymptomatic, this means the number of active symptomatic cases on Day 14 after lifting lockdown restrictions would have increased to about 255 and the number of new symptomatic cases would be about 9.

## Discussion

Our findings suggest that the multiple control measures adopted in Sri Lanka during March 2020, which includes prompt contact tracing and isolation, border closure and complete lockdown, have enabled reduction in transmission from an initial level (R = 3.0) that would have almost certainly overwhelmed Sri Lanka's health system within a month, peaking in about 3 months, with well over 5 million active infections at that point.

The simple SIR model we developed enables visualization of how different levels of control would affect the speed at which ICU capacity in our country reaches saturation and the number of cases that would signal the likelihood of this occurring in 2–3 months. Our projections suggest that transmission should be controlled so that R does not exceed 1.5 for any prolonged length of time, in order to avoid overloading the ICU capacity. The model can also be used to envisage the impact of varying levels of control in different areas within Sri Lanka, such as in the 6 districts in Sri Lanka categorized as having a high risk of transmission compared to the other 19 districts which have a lower risk. This could inform healthcare decision making at a more local level.

It may be argued that the SIR model is not applicable in the Sri Lankan context, because there is, as yet, no full-blown community transmission. However, it is likely that the COVID-19 epidemic in Sri Lanka will move into this phase, as has happened in many other countries over the past three months, and the SIR model is widely accepted as a means of conceptualizing the spread of an infectious disease through a population over time [10].

The validity of the projections derived from our model depend a great deal on the accuracy of the assumed parameters, such as the proportion of identified cases, the average period of infectiousness, the proportion of individuals who require ICU care, the duration of ICU stay, etc. The estimates presented here are based on data reported from other countries where the epidemic is more advanced, and may not necessarily be appropriate in the Sri Lankan context. However, the availability of the app enables the user to change the parameters as required as more data becomes available.

Other web-based applications have been developed, such as the Epidemic Calculator available at https://gabgoh.github.io/COVID/index.html. This application uses a SEIR (Susceptible, Exposed, Infectious, Removed) model, and although it does not enable calculation of the saturation of ICU bed capacity, the results produced by our model in terms of active infections, susceptible individuals and recovered patients are on par with the Epidemic Calculator under no intervention scenario. We chose not to use a SEIR model because the data available at this stage in Sri Lanka was insufficient to estimate all the parameters necessary for such a model.

Another web-based app developed at the London School of Hygiene & Tropical Medicine is available at https://cmmid-lshtm.shinyapps.io/hospital_bed_occupancy_projections/, to estimate projected hospital bed occupancy in the UK. However, this app can be used to forecast COVID-19 bed requirements in a given location for only up to 21 days (e.g. a healthcare facility, a county, a state) and our estimates were similar to this app.

We believe that the model and web based app, which we developed primarily for use in Sri Lanka, may also be appropriate for use in other low and middle income countries that have similar constraints for ICU care of COVID-19 patients, but are unable to enforce stringent lockdown measures for a prolonged period of time due to social and economic reasons.

## Supporting information

**S1 Data. Covid19 cases SL.**
(XLSX)

## Acknowledgments

We thank Prof Deirdre Hollingsworth, Prof Don Bundy, Prof Rajitha Wickremasinghe and Dr Sudath Samaraweera for helpful guidance and comments on the draft manuscript and Dr Prasad Ranatunga for helpful comments on the web-based application.

## Author Contributions

**Conceptualization:** Dileepa Senajith Ediriweera, Nilanthi Renuka de Silva, Hithanadura Janaka de Silva.

**Data curation:** Dileepa Senajith Ediriweera.

**Formal analysis:** Dileepa Senajith Ediriweera.

**Investigation:** Dileepa Senajith Ediriweera, Nilanthi Renuka de Silva.

**Methodology:** Dileepa Senajith Ediriweera, Nilanthi Renuka de Silva.

**Software:** Dileepa Senajith Ediriweera.

**Supervision:** Nilanthi Renuka de Silva, Gathsaurie Neelika Malavige, Hithanadura Janaka de Silva.

**Validation:** Dileepa Senajith Ediriweera, Nilanthi Renuka de Silva.

**Visualization:** Dileepa Senajith Ediriweera, Nilanthi Renuka de Silva, Hithanadura Janaka de Silva.

**Writing – original draft:** Dileepa Senajith Ediriweera, Nilanthi Renuka de Silva, Gathsaurie Neelika Malavige, Hithanadura Janaka de Silva.

**Writing – review & editing:** Dileepa Senajith Ediriweera, Nilanthi Renuka de Silva, Gathsaurie Neelika Malavige, Hithanadura Janaka de Silva.

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
