## [Decision Letter · Decision Letter 0]

2 Aug 2020

PONE-D-20-11682

AN EPIDEMIOLOGICAL MODEL TO AID DECISION-MAKING FOR COVID-19 CONTROL IN SRI LANKA

PLOS ONE

Dear Dr. Ediriweera,

Thank you for submitting your manuscript to PLOS ONE. After careful consideration, we feel that it has merit but does not fully meet PLOS ONE’s publication criteria as it currently stands. Therefore, we invite you to submit a revised version of the manuscript that addresses the points raised during the review process.

In their manuscript the Authors provide an interesting tool. They should better clarify how this tool can be used in different countries (i.e. low countries). 

We look forward to receiving your revised manuscript.

Kind regards,

Chiara Lazzeri

Academic Editor

PLOS ONE

Journal Requirements:

2. Please clarify the term 'lockout' with a definition.

3.We note that the figuresin your submission contain copyrighted images. All PLOS content is published under the Creative Commons Attribution License (CC BY 4.0), which means that the manuscript, images, and Supporting Information files will be freely available online, and any third party is permitted to access, download, copy, distribute, and use these materials in any way, even commercially, with proper attribution. For more information, see our copyright guidelines: http://journals.plos.org/plosone/s/licenses-and-copyright.

1.    You may seek permission from the original copyright holder of the figures to publish the content specifically under the CC BY 4.0 license.

Reviewers' comments:

Reviewer's Responses to Questions

**Comments to the Author**

1. Is the manuscript technically sound, and do the data support the conclusions?

Reviewer #1: Yes

2. Has the statistical analysis been performed appropriately and rigorously? 

Reviewer #1: I Don't Know

3. Have the authors made all data underlying the findings in their manuscript fully available?

Reviewer #1: No

4. Is the manuscript presented in an intelligible fashion and written in standard English?

Reviewer #1: Yes

5. Review Comments to the Author

Reviewer #1: The manuscript is interesting and seems to provide value as a unique model to predict ICU bed needs based on the R naught. However, I find the wording "Active symptomatic cases in the community:" on the model to be confusing. Perhaps the authors could re-title the section to something more clear. Asymptomatic or pre-symptomatic individuals can spread the virus, so it is confusing as to why only "symptomatic" cases would be included as a factor in predictions. I assume this may be that in Sri Lanka, at the time of writing, only symptomatic individuals were being tested. Either way, I think the wording could be more clear so that those using the model can better understand the metrics that affect the output.

This manuscript does demonstrate the importance of social distancing and other infection prevention measures to maintain ICU bed availability. I think this could be particularly helpful to communities that have had little had SARS-CoV-2 spread in their communities in helping them determine what social distancing measures to keep their R naught values low. It is likely that countries with little COVID-19 diagnoses to date, particularly those with less resources, would find this modeling tool of particular use.

6. PLOS authors have the option to publish the peer review history of their article (what does this mean?). If published, this will include your full peer review and any attached files.

Reviewer #1: No

---

## [Author Response · Author response to Decision Letter 0]

5 Aug 2020

Editors comments:

• Done. 

2. Please clarify the term 'lockout' with a definition.

• We have now replaced lockout with “lifting lockdown restrictions”. 

3.We note that the figures in your submission contain copyrighted images. All PLOS content is published under the Creative Commons Attribution License (CC BY 4.0), which means that the manuscript, images, and Supporting Information files will be freely available online, and any third party is permitted to access, download, copy, distribute, and use these materials in any way, even commercially, with proper attribution. For more information, see our copyright guidelines: http://journals.plos.org/plosone/s/licenses-and-copyright.

• We have not used any copyrighted images in this manuscript and all the images in the manuscript were developed by us from the publicly available data. 

Reviewer #1: The manuscript is interesting and seems to provide value as a unique model to predict ICU bed needs based on the R naught. However, I find the wording "Active symptomatic cases in the community:" on the model to be confusing. Perhaps the authors could re-title the section to something more clear. Asymptomatic or pre-symptomatic individuals can spread the virus, so it is confusing as to why only "symptomatic" cases would be included as a factor in predictions. I assume this may be that in Sri Lanka, at the time of writing, only symptomatic individuals were being tested. Either way, I think the wording could be more clear so that those using the model can better understand the metrics that affect the output.

• Here, we obtain both “Active symptomatic cases in the community:” and “Percentage of symptomatic cases out of total infected (%)” in the application. 

• We calculate the total infections as “Active symptomatic cases in the community:” divide by “Percentage of symptomatic cases out of total infected (%)”. 

• At the time of the manuscript writing, only the symptomatic individuals clinically suspected of infection with SARS-CoV-2 were tested with rt-PCR and diagnosed. 

• We agree that we can re-title these as below;

o “Active symptomatic cases in the community” as “Number of identified cases up to now:”

o “Percentage of symptomatic cases out of total infected (%)” as “Percentage of identified cases out of total infected (%)” 

This manuscript does demonstrate the importance of social distancing and other infection prevention measures to maintain ICU bed availability. I think this could be particularly helpful to communities that have had little had SARS-CoV-2 spread in their communities in helping them determine what social distancing measures to keep their R naught values low. It is likely that countries with little COVID-19 diagnoses to date, particularly those with less resources, would find this modeling tool of particular use.

• Thank you for your comment.

---

## [Editor Report · Decision Letter 1]

17 Aug 2020

AN EPIDEMIOLOGICAL MODEL TO AID DECISION-MAKING FOR COVID-19 CONTROL IN SRI LANKA

PONE-D-20-11682R1

Dear Dr. Ediriweera,

We’re pleased to inform you that your manuscript has been judged scientifically suitable for publication and will be formally accepted for publication once it meets all outstanding technical requirements.

Kind regards,

Chiara Lazzeri

Academic Editor

PLOS ONE
---

## [Editor Report · Acceptance letter]

19 Aug 2020

PONE-D-20-11682R1 

An epidemiological model to aid decision-making for covid-19 control in Sri Lanka 

Dear Dr. Ediriweera:

I'm pleased to inform you that your manuscript has been deemed suitable for publication in PLOS ONE. Congratulations! Your manuscript is now with our production department. 

Kind regards, 

on behalf of

Dr. Chiara Lazzeri 

Academic Editor

PLOS ONE